# Standardized Description of Degraded Land Reclamation Actions and Mapping of Actors' Roles: A Key Step for Action in Combatting Desertification (Niger)

**Abou-Soufianou Sadda** [1], **Maud Loireau** [2], **Nouhou Salifou Jangorzo** [3], **Hassane Bil-Assanou Issoufou** [3] and **Jean-Luc Chotte** [4,*]

1   Institut de Recherche pour le Développement, Représentation, 276 Avenue de Maradi, Niamey BP 11 416, Niger; abou-soufianou.sadda@ird.fr
2   Institut de Recherche pour le Développement, Espace-DEV, Universités Montpellier, de la Réunion, de Guyane, des Antilles, de Perpignan, 66860 Perpignan, France; maud.loireau@ird.fr
3   UMR DAP, Université Dan Dicko Dankoulodo de Maradi, Maradi BP 465, Niger; nouhou.jangorzo@gmail.com (N.S.J.); hassanebil-assanou.issoufou@ird.fr (H.B.-A.I.)
4   Institut de Recherche pour le Développement, UMR Eco&Sols, Université Montpellier, IRD, CIRAD, INRAE, SupAgro, 34060 Montpellier, France
*   Correspondence: jean-luc.chotte@ird.fr

**Abstract:** Land degradation is a major issue in the Sahel region. Numerous investments have been made in implementing sustainable land management (SLM) actions to reverse land degradation. Our work aims to (i) describe the variety of degraded land reclamation actions (DLRAs) and (ii) map the stakeholders acting in Niger. A time series (2008–2021) of georeferenced public data was collected and organized using a harmonized nomenclature. The results show that about 279,074 ha could be analysed in our study. Dug structures are the most widespread technique, while treated land is mostly devoted to single agricultural or pastoral uses. DLRAs are unevenly distributed in the Niger. More than 100 stakeholders were part of the effort to restore degraded land in the country—some playing a specific role, while others, such as the Government of the Niger, were responsible for mobilizing funds for implementing sustainable land management programs, while also carrying out certain programmes of their own. Our study points out the added value of creating a geolocalized dataset and, in future, a spatialized database management system to (i) deploy targeted sustainable land management actions complementing past and ongoing actions and (ii) create synergy between all the stakeholders.

**Keywords:** biophysical actions; combat land degradation; stakeholder network; spatiotemporal database; traceability; monitoring and evaluation





## 1. Introduction

The well-being and livelihoods of rural populations are strongly dependent on the health and productivity of land [1,2]. Natural resources overuse and increasing demand are causing rapid land degradation worldwide [3]. Degradation is characterized by a negative trend in the state of the land [4]. It involves the total or partial loss of vegetation cover, soil fertility, productivity, and/or biodiversity, leading to a decline in ecosystem services and socio-ecosystem resilience [5]. In fact, 52% of soils are moderately or severely degraded on a global scale [3]. In sub-Saharan Africa, the situation is worse. It is estimated that 75% of arable land is degraded or highly degraded [6].

In the Sahel, land degradation is the result of human activities that overexploit non-renewable natural resources in a constrained biophysical environment [7–9], exacerbated by climate change and biodiversity loss [10]. In most Sahelian countries, land degradation is not compensated by actions that aim to restore or rehabilitate degraded land. The net result is negative [11]. Thus, the deterioration of the physical-chemical and hydrological

properties of soils leads to an increase in water and wind erosion [12], and a decrease in the productive capacity of semi-arid ecosystems. Economic losses equivalent to 10–17% of global GDP are attributed to land degradation [13]. The well-being of 3.2 billion people is impacted, and one million animal and plant species could disappear by 2050 [14].

Reducing or slowing down land degradation, and rehabilitating or restoring (where possible) degraded land is the challenge of Sustainable Development Goal (SDG) 15.3 of the United Nations. Moreover, beyond this single target, this issue also concerns food security (SDG 2), poverty reduction (SDG 1), water quality (SDG 6), and mitigation of and adaptation to climate change [15].

Several international initiatives promoting actions to treat degraded land have been launched. These include the Bonn Challenge launched in 2011 with the goal of treating 150 million ha by 2020 and 350 million ha by 2030 [16]. The African Forest Landscape Restoration Initiative launched in 2015 aims to treat 100 million ha of degraded land by 2030 [16]. The "4‰ Initiative: Soils for Food Security and Climate" launched in 2015 proposes to annually increase the organic carbon stock in cultivated soils by 4‰ at the global level to offset the annual increase in anthropogenic $CO_2$ emissions. In the Sahel, the Great Green Wall (GGW) initiative launched in 2007 aims to treat 100 million ha of degraded land by 2030.

In this context, adopting sustainable land management practices (SLM) is a solution to promote better management of natural resources and establish the foundations of sustainable economic and social development [17]. SLM practices and approaches help treat degraded land [18,19], but also prevent or slow down their degradation.

As in other Sahelian countries, in the Niger the strong anthropization of rural areas has led to the degradation of over 60 percent of arable land [20]. This situation is characterized by the disappearance of vegetation cover and soil crusting [21–23]. Many SLM initiatives have been implemented since 1984, with combating desertification declared a "national priority." As a matter of fact, projects and programmes led by a wide diversity of stakeholders have endeavoured to treat land degradation by promoting a wide range of techniques and technologies. However, there has been a lack of stocktaking at the national level in the Niger around the implementation of degradation land reclamation actions—both in terms of location and the roles played by the various stakeholders. Our work, part of a broader project entitled "Large scale assessment of land degradation to guide future investments in sustainable land management in the Great Green Wall countries (Global Environment Facility grant 9825), aims to fill those gaps. In addition, it maps the links between the roles of the different actors (donors, fund mobilizers, operators, and implementers). To this end, we have developed a spatially referenced data table that lists and maps temporally sequenced (2008–2021) geolocated data. These data have been collected from publicly accessible sources.

## 2. Methodological Approaches

*2.1. Data Collection on SLM Actions and Construction of a Spatially Referenced Data Table, Viewable in GIS, for the Traceability of Degraded Land Reclamation Actions (DLRA)*

Seven main steps were followed (Figure 1).

Step 1: Information mobilization

The aim was to collect and centralize all types of public data dealing with SLM actions carried out in Niger between February and June 2022, from the oldest to the most recent, in various formats and media, available online or distributed among the structures holding the data (public, parapublic, or private). To identify these different structures, a pre-established survey form was distributed to an initial list of 20 active structures, made up of NGOs, projects, and development programmes present in Niger. Interviews were also conducted with civil society and bilateral cooperation actors in order to make a brief diagnosis of the databases of SLM actions in Niger. The consultation of technical documents (such as study reports), scientific articles, websites, and institutional databases dealing with degradation, treatment, or SLM, helped complete the information thus collected.

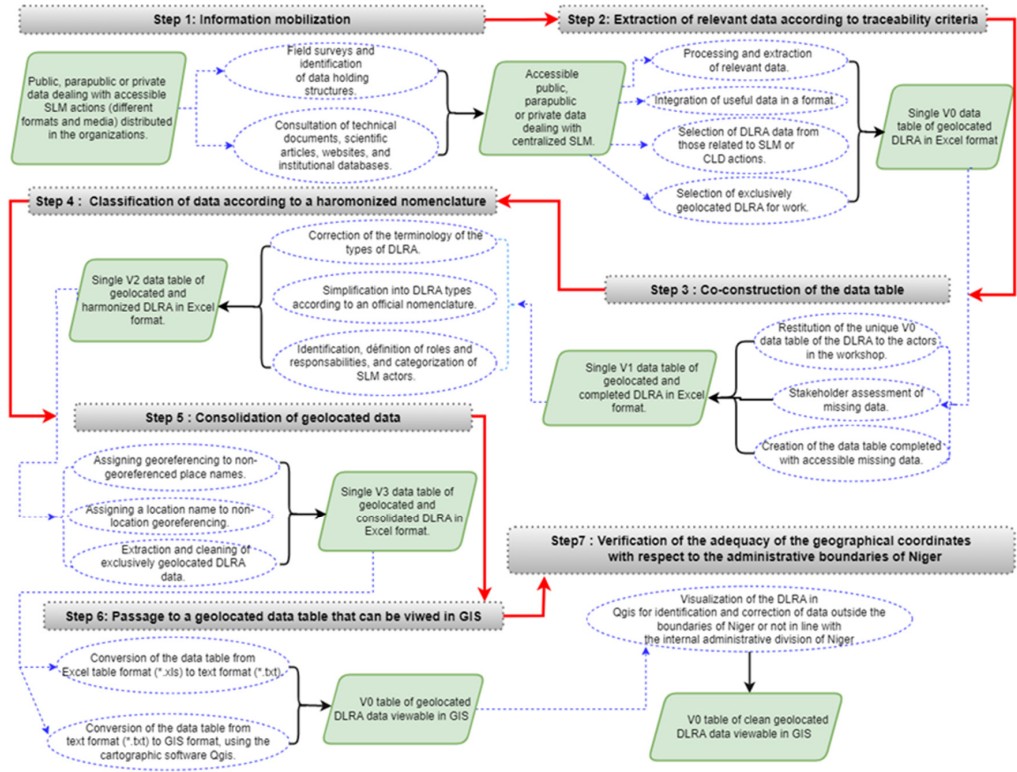

**Figure 1.** Steps to build a GIS of DLRA data according to traceability criteria. Note: V0 = Version 0; V1 = Version 1; V2 = Version 2; V3 = Version 3.

Step 2: Extraction of relevant data according to traceability criteria

The information collected was sorted, and the useful data were extracted and integrated into a single data table in Excel format. From the range of SLM actions or those designed to combat land degradation (ADA), the DLRA implemented in the field were selected (Figure 2).

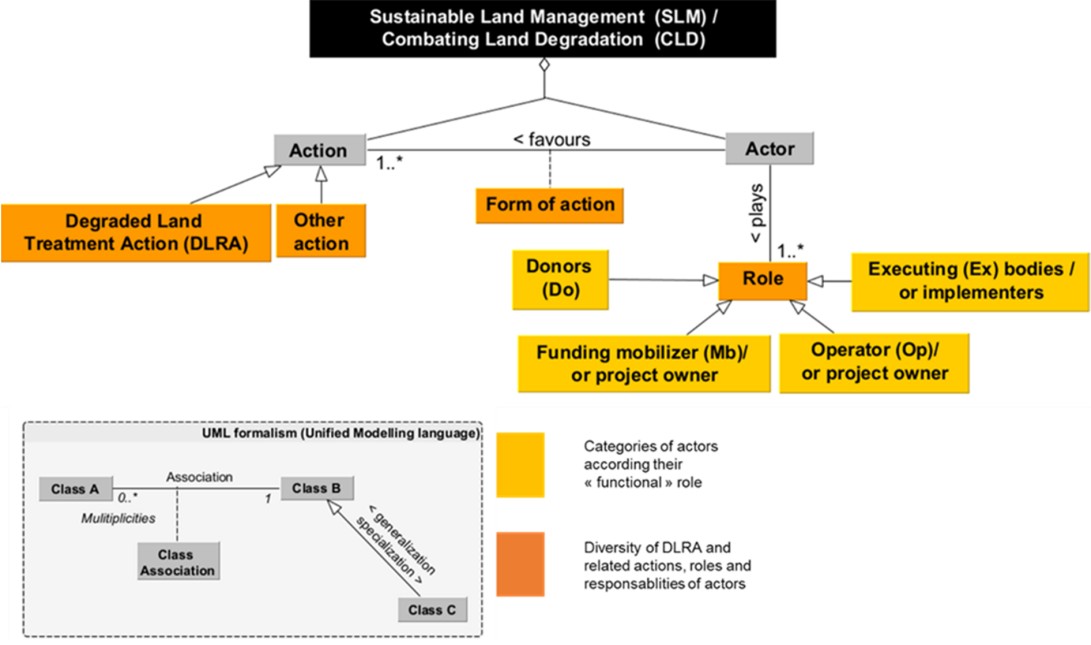

**Figure 2.** Schematic of the method for classifying DLRA data according to a harmonized nomenclature. Note: * indicates that an actor can play one or several roles in SLM in Niger.

Only geolocated DLRAs were selected for our work. Geolocation is a key traceability criterion, as it allows anyone to return to the places where the DLRAs were implemented and to avoid having to inventory the same action several times. The other traceability criteria are the type of DLRA; the area treated, the spatial entity (region, department, commune, village), the year, the purpose; and the donors, the actors who mobilize these donors, the operators, and the implementing entities on the ground.

Step 3: Co-construction of the data table

The result of this work of collecting and organizing data on DLRAs was presented to some thirty actors representing the various structures in order to gather their analysis on the construction of this data table and to enrich it with projects that had been overlooked until then.

Step 4: Classification of data according to a harmonized nomenclature

Each DLRA was described according to the official national technical sheets for SLM actions in Niger [24,25]. When the DLRA at a site is a combination of several actions, the name of the action with the largest treated area is used.

To define the roles and responsibilities of the different actors, it was necessary to return to the project documents. Four categories of actors were identified along the chain (Figure 2): (1) donors (Do); (2) fund mobilizers (Fm)—who identify and mobilize funds; (3) operators (Op)—who put DLRAs in action; and (4) and implementers (Im)—who implement DRFLAs in the field.

Step 5: Consolidation of geolocated data

The collection of data from multiple sources generated geolocation errors and inconsistencies. Systematic georeferencing and localization work was then carried out: (1) by assigning a georeference to the names of non-georeferenced localities, and conversely, (2) by assigning the name of a locality to the georeference without locality. All these operations were carried out using tools such as Google Earth and Geonames (https://www.geonames.org/, accessed on 12 December 2022). Only those DLRAs that were consolidated in this way from a geolocation point of view were retained. The subsequent cleaning of the data table consisted of (1) deleting the data collected that did not relate to the DLRA; (2) deleting the DLRAs that did not contain all the descriptive data that would allow them to be traced (cf. step 2); (3) correcting the input errors detected and standardizing the formats of the geographical coordinates; and (4) deleting the duplicates.

Step 6: Passage to a geolocated data table that can be viewed in GIS

Once the geolocated dataset and other traceability criteria had been filled in, their Excel table format (*.xls) was converted into text format (*.txt) and then into GIS format, using the mapping software Qgis.

Step 7: Verification of the adequacy of the geographical coordinates with respect to the administrative boundaries of Niger

Viewing the DLRAs in Qgis revealed that some of them are outside Niger or do not fit into the internal administrative division (region, department, commune). Using the attribute table and tools such as Google Earth and Geonames, they were then brought back to the corresponding administrative boundaries.

### 2.2. Data Analysis

The analytical work is based on a single parameter: the total area of land treated contained in our date base (e.g., 279,074 ha). The analyses (in Excel 2013) were carried out on the harmonized and consolidated DLRA data. The e!Sankey software (version 5.2.1) was used to visualize (i) for each of the roles (e.g., Do, Fm, Op, and Im), the respective contribution to each of the categories of actors (see Tables S1–S5); and (ii) their interactions (e.g., the funds of which donors are mobilized by which fund mobilizers, and then used to support the actions of which operators, who in turn entrust which category of implementers to carry out the actions required to treat a total of 279,074 ha). The size of the nodes (Do, Fm, Op, and Im) and the links between each category of actors depend on their contribution to

the role and the interactions of the actors. The results are expressed as a percentage of the total area treated (Table S6).

## 3. Results

### 3.1. Data Table: Keys to Identifying and Harmonizing the Nomenclature of DLRA in Niger

Given the diversity of DLRAs in terms of an official nomenclature (see Step 4), they have been grouped together according to two levels of structuring (Figure 3: in green, first level; in grey, second level). This grouping is the result of a consensus between Nigerien LCD experts at the end of a three-day workshop.

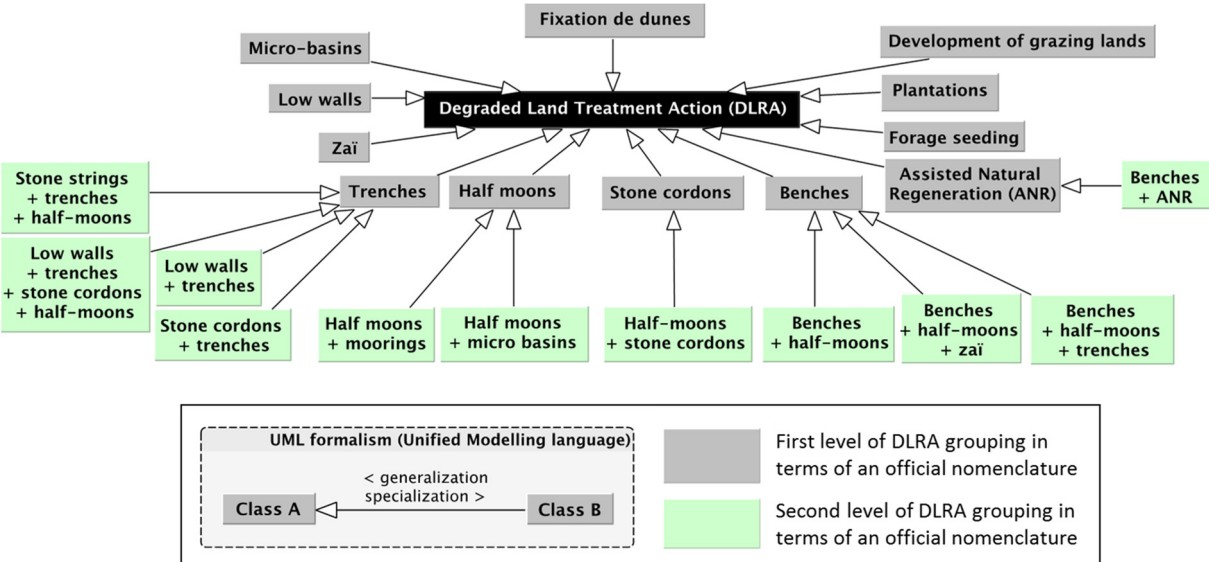

**Figure 3.** Scheme for grouping DLRAs in Niger according to a harmonized nomenclature.

Trenches are works dug along a line, within which several stone cordons may be placed, and between two lines of trenches several half-moons are dug. In principle, where trenches exist, they take precedence over other excavated works. The same applies to a low wall that is dug along a slope or a gully. In a trenched area, one or more low walls can be built upstream to slow down the runoff.

The DLRAs grouped under the nomenclature of "half-moons" are the combinations of half-moons associated with low walls and half-moons associated with micro-pools. The half-moons therefore take precedence over the low walls and micro-pools constructed to treat koris. A kori is a low-lying area with the appearance of a dry riverbed in the dry season and experiencing strong runoff after the rain. The low walls and micro-pools are most often downstream of other dug structures, such as half-moons, that occupy a higher surface.

The DLRAs grouped under the nomenclature of "stone cordons" are in fact half-moons associated with stone cordons, based on the principle that between two lines of stone cordons several half-moons can be dug.

The DLRAs grouped under the nomenclature of "benches" are those associated with half-moons or zaïs, or those associated with trenches. In fact, benches are long excavations laid out on the contour line. They consist of a bead at the downstream end and a ditch at the upstream end with two wings, where the space between the wings and along the edges can be used to dig half-moons, zaïs, or trenches.

In total, 17 "elementary" DLRA types were differentiated, but as the typology in Figure 3 shows, they are most often combined on the same plot:

- Eleven of these are dug structures (half-moons, benches, stone barriers, trenches, low walls, zaï, and micro-basins) for soil and water conservation (SWC), soil defence, and restoration (SDR), and almost always combined; these are mechanical actions.
- One is a biological action of SWC and SDR, namely dune fixation.
- Four are biological agricultural actions (assisted natural regeneration (ANR) practices and tree planting in agroforestry/forestry and pastoral rangeland management and forage seeding in pastoralism).
- A combination of mechanical (bench) and biological (ANR) actions.

### 3.2. Dugouts at the Heart of Degraded Land Treatment in Niger

The DLRA data table shows that a total of 279,074 ha of degraded land treated in Niger has been georeferenced over the period 2008 to 2021. In general, dug-out structures remain at 80% (i.e., 223,822 ha of the total area of treated land) at the heart of the treatment of degraded land in Niger. However, the georeferenced treated areas vary significantly between the different types of DLRA (Figure 4). Thus, with 170,424 ha (61%), half-moons occupy the largest area of total land treated. They are followed by dune fixation and benches with, respectively, 39,230 ha (14%) and 38,072 ha (13%). The other DLRA represent only 12% of the total area treated, with 7806 ha (3%) for stone cordons, 7394 ha for forage seeding (3%), 4534 ha (2%) for trenches, 3122 ha (1%) for plantations, 3090 ha (1%) for ANR, 2415 ha (0.9%) for grazing lands, 1398 ha (0.6%) for zaï, 1173 ha (0.4%) for low walls, and 414 ha (0.1%) for micro-basins.

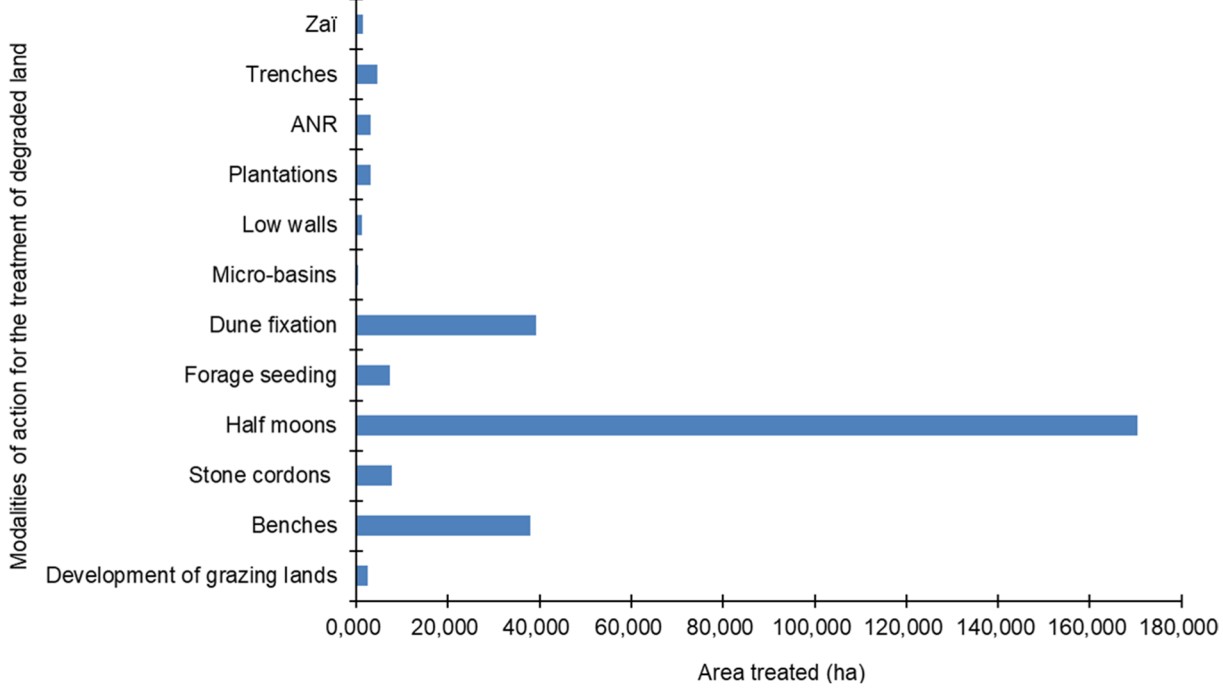

**Figure 4.** Distribution of the area of degraded land treated according to the type of DLRA carried out in Niger.

The DLRA are implemented on land with different uses (Figure 5). The largest proportion of land treated (168,992 ha, or 60%) is agricultural or pastoral. Pastoral land (e.g., silvopastoral, agropastoral, and agrosylvopastoral) and forestry land (silvicultural and agrosilvicultural) represent 36% and 4% of the land treated, respectively.

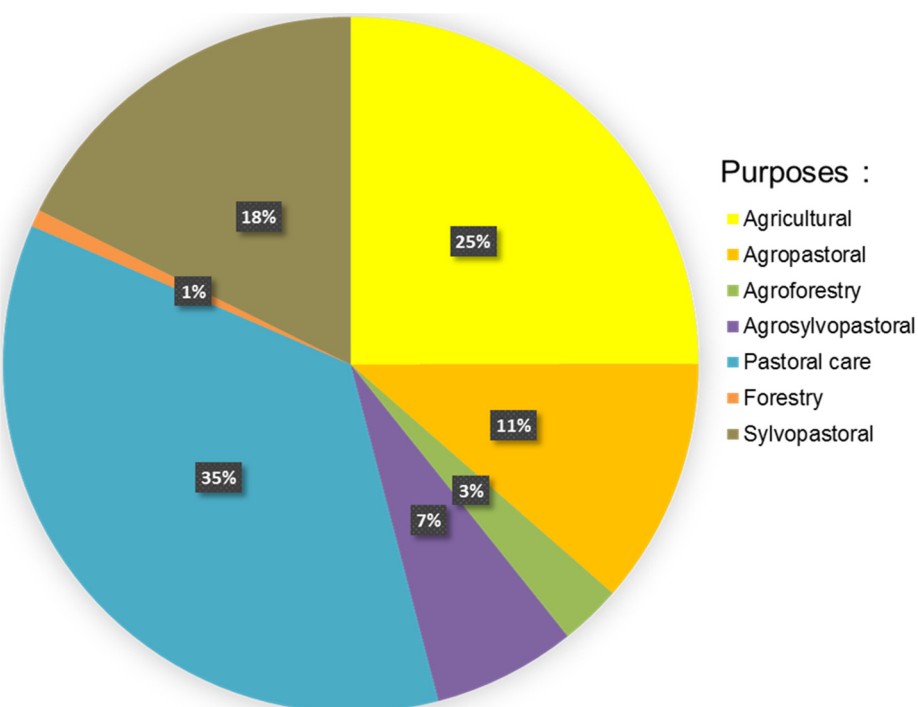

**Figure 5.** Breakdown of the proportion of degraded land treated by type of DLRA use provided in Niger.

### 3.3. DLRAs Are Unevenly Distributed in Niger

3.3.1. Analysis by Administrative Region of the Spatial Distribution of DLRAs

From 2008 to 2021, the proportions of land treated in each year relative to the total over the period vary from year to year (Figure 6). In 2012, the most land was treated by DLRA. Indeed, almost 20% of all land treated was treated in 2012, that is 50,844 ha. In 2013, 2014, 2015, 2016, 2017, and 2021, about 10% of the total 279,074 ha recorded in our database was treated per year. The years 2008, 2009, 2010, 2011, 2018, 2019, and 2021 saw low investments in SLM. The political will of the State of Niger is illustrated through the launch of the 3N programme "Nigériens Nourrissent les Nigériens", and the implementation of various programmes (e.g., COMPACT of the Millennium Challenge Account, the Programme for the Development of Family Farming in the Regions of Maradi, Tahoua and Zinder (ProDAF), and the Support Project for Rural Activities and Financing of Agricultural Commodity Chains in the regions of Agadez and Tahoua). On the other hand, the years of low investment can be explained both by projects that have come to an end, others that have not started, and for 2020 by the health situation slowing down the execution of the projects.

Expressed as a proportion of the total area of land treated (i.e., 279,074 ha), the contribution of the actions differs according to the administrative regions of Niger (Figure 7). The Tahoua region represents, with all types of actions taken together, the highest proportion of the total land treated (26%). The existence, since 2008, of data documenting the actions undertaken in this region is one of the reasons for this result. In contrast, a lower share of land treatment actions in the other regions can be explained by a lack of data. Conversely, only 1% of the actions to treat degraded land are carried out in the Niamey region. Without ignoring the issue of archiving actions, this result can also be explained by the very urban nature of this region and therefore the weakness of efforts made on the outskirts of the city. For the other regions, the proportions of land treated are almost equivalent, in ascending order, between the regions of Zinder (14%) and Diffa (13%), Maradi (16%) and Tillabéri (16%), and finally Dosso (7%) and Agadez (7%).

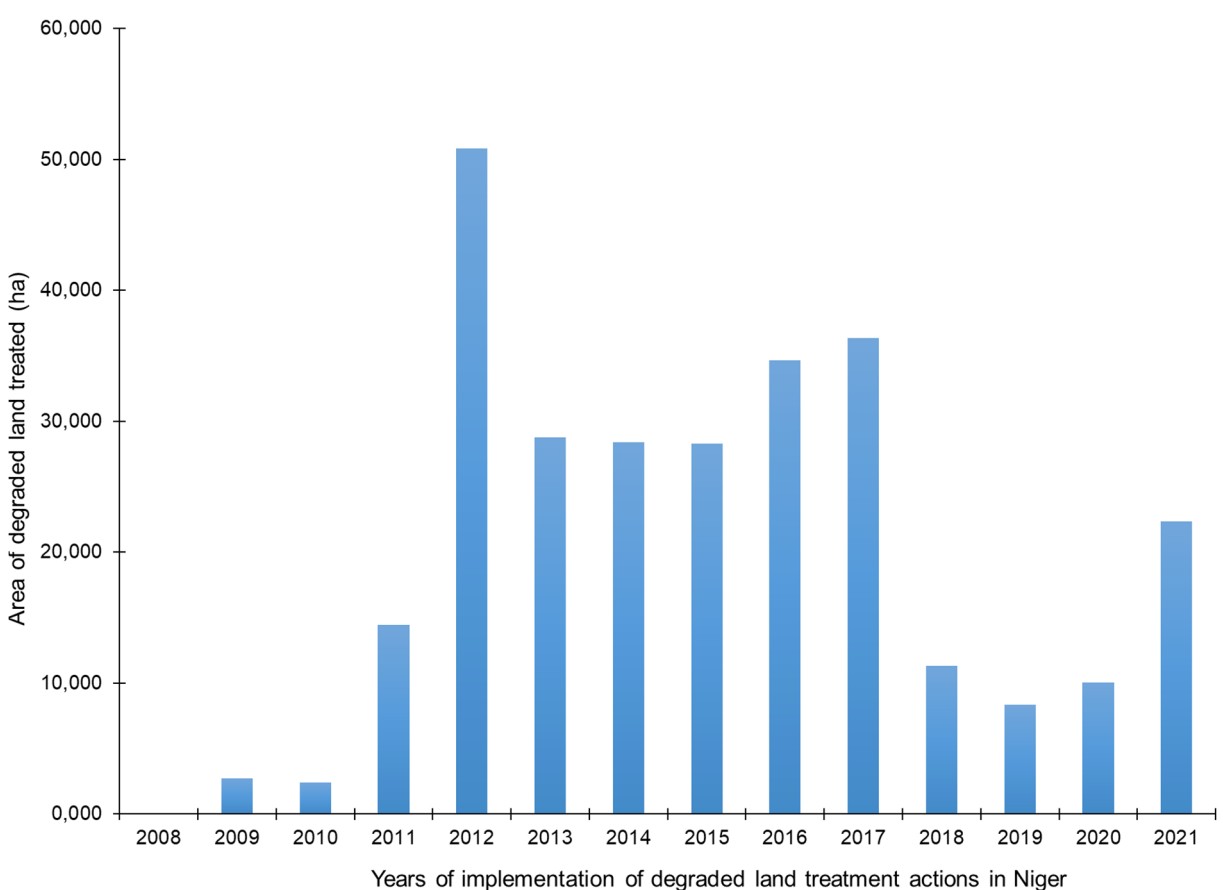

**Figure 6.** Variation in area of DLRAs by year in Niger.

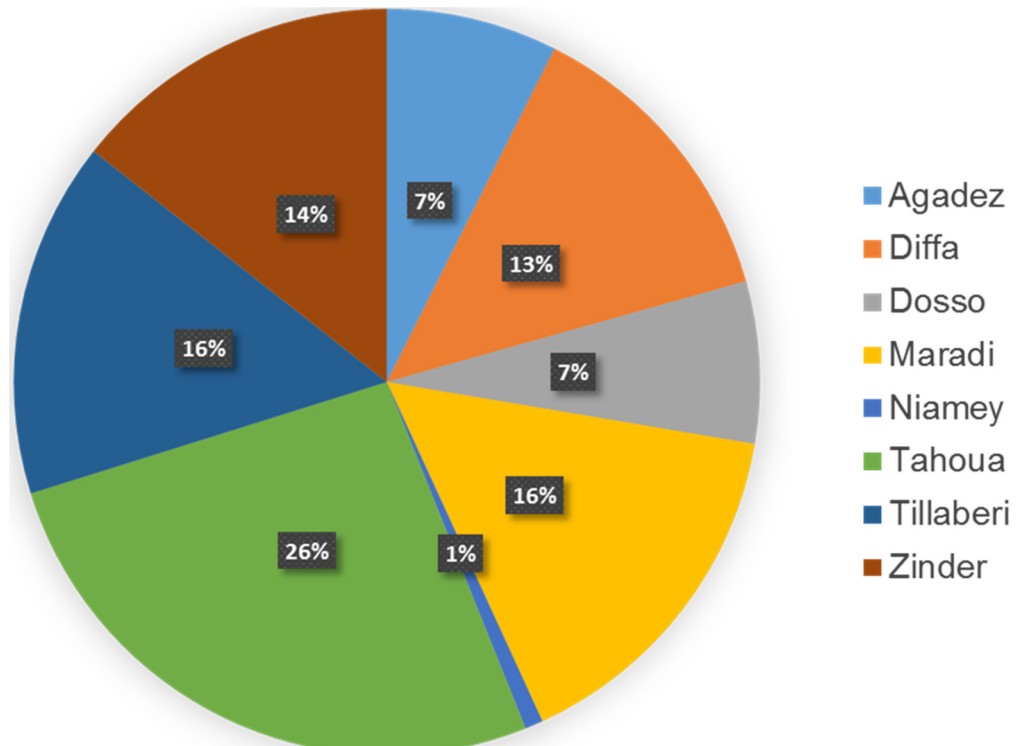

**Figure 7.** Distribution of the proportion of degraded land treated according to administrative regions in Niger.

3.3.2. Type Analysis of the Spatial Distribution of DLRA

The visual analysis is made possible by the use of symbols in Figure 8 that distinguish between types of DLRA. Geolocated DLRAs are found mainly in the southern half of Niger, an agricultural area par excellence. Half-moons and benches predominate in all regions of Niger, except in Diffa where benches are absent. The sandy and sometimes clayey soils in the lowlands of this region are not conducive to the deployment of benches.

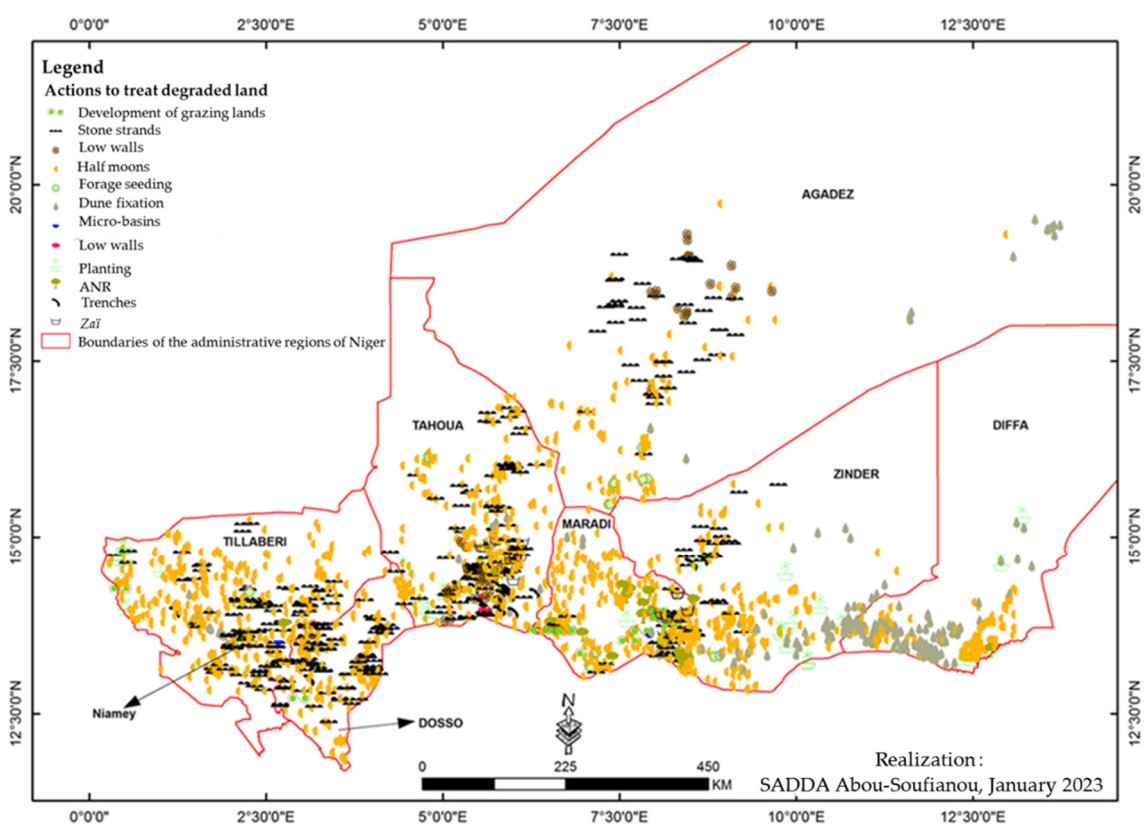

**Figure 8.** Spatial distribution of geolocated DLRAs in Niger.

The practice of dune fixation is mainly observed in the regions of Diffa, Tillabéri, and Zinder, more marginally in the south-west and south-east of the Agadez region, in the north and east of the Maradi region, and in the east of the Tahoua region. These are the northernmost parts of the country from east to west, most affected by wind erosion leading to sand dune movements. These are also pastoral areas par excellence where pressure on plant resources is very intense, leading to soil denudation. RNA and tree planting practices are widespread, except in the regions of Dosso, Niamey, and Agadez. Niamey is a highly urbanized area not oriented towards rural agricultural activities, while much of the Dosso area is either covered by in-ground plateaus or covered by woody vegetation (W park forest). Fodder sowing is observed everywhere except in the regions of Dosso, Niamey, and Diffa. Dosso and Niamey do not constitute a livestock zone par excellence, and therefore do not have degraded pastoral land. The size of the pastoral area in the Diffa region has probably limited seeding efforts.

Some DLRAs are implemented exclusively in certain regions: low walls in the Tahoua region; trenches in the Tahoua, Zinder, Dosso, and Tillabéri regions; stone barriers in the Agadez, Tahoua, Dosso, and Niamey regions; micro-basins in the Maradi and Tillabéri regions; zaïs in the Maradi and Tahoua regions; and pastoral rangeland development in the Tillabéri, Maradi, and Dosso regions (Figure 8). This specialization is linked to the geomorphological and agroecological characteristics of the regions but also to the requirements of the practices themselves. For example, the stone cordons require the

presence of stone, a resource which the regions located on the plateaus and plinths are endowed with, as is the case in Tahoua, Agadez, Dosso, and Tillabéri.

The micro-basins are constructed in rainy areas with relief (Tillabéri) or soils sensitive to linear erosion that can create koris and gullies (Maradi). The zaï are more suitable for DLRAs on the crested soils of agricultural areas (Tahoua) or to optimize organic fertilization on degraded sandy soils (Maradi).

### 3.4. Mapping the Actors' Role: The State of Niger, a Key Player in the Fight against Land Degradation in Niger

The contributions of the more than 100 stakeholders identified in our database to the 279,074 ha processed by DLRAs can be distinguished according to their "functional" role: (1) as donors (Do), (2) as fund mobilizers (Fm), (3) as operators (Op), and (4) as implementers (Figure 9).

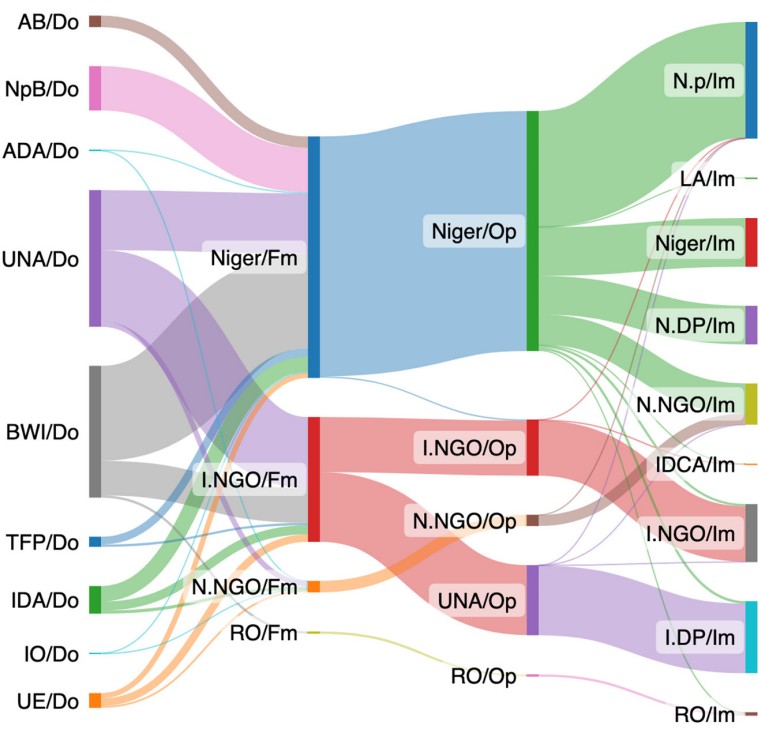

**Figure 9.** Chain of actors for sustainable land management in Niger. Note: left to right: (1) donors (Do); (2) fund mobilizers (Fm), i.e., an actor who identifies and mobilizes funds; (3) operators (Op), i.e., an actor who puts DLRA in action; and (4) implementers (Im), i.e., an actor who implements DLRAs in the field. Legend: Donors: ADA/Do—African development agencies; IDA/Do— international development agencies; UNA/Do—United Nations' agencies as a donor; AB/Do—African Bank; CB/Do—community banks; NpB/Do—The Niger public budget; BWI/Do—Bretton Woods Institutions; IO/Do—international organizations; TFP/Do—technical and financial partners; EU/ Do—European Union (See detail in Table S1). Fund Mobilizers: Niger/Fm—The Niger public authority; I.NGO/Fm—NGOs and international associations; N.NGO/Fm—NGOs and Nigerien associations; RO/Fm—sub-regional organizations (See detail in Table S2). Operators: UNA/Op—United Nations' Agencies; Niger/Op—The Niger as a fundraiser; LA/Op—local authorities; I.NGO/Op— NGOs and international associations; N.NGO/Op—NGOs and Nigerien associations; RO/Op—sub-regional organizations (see detail in Table S4). Implementers: IDCA/Im—International development cooperation agencies; LA/Im—local authorities as an implementer; Niger/Im—The Niger as an implementer; I.NGO/Im—NGOs and international associations as an implementer; N-NGO/Im—NGOs and Nigerien associations; I.DP/Im—international development programmes (see detail in Table S5).

Ten categories of donors financed the implementation of DLRAs in Niger (Table S1) and for various areas (Figure 9). The UN agencies and Bretton Wood institutions are the main donors, whose financing represents, respectively, 36% (100,565 ha) and 34% (96,825 ha) of the total area of DLRAs. The Government of Niger, with its own funds, comes in third place with 12% (35,716 ha) of DLRAs. Other donors (e.g., international agencies, the European Union, African banks, technical and financial partners, African development agencies, community banks, and international organizations) have made funding available to treat 7%, 4%, 3%, and 1%, respectively, of the total area of land treated in Niger.

As an actor involved in the mobilization of funding (Table S2), the Government of Niger plays a key role. Its action has made it possible to treat 63% of the total area. In this role of mobilizing funds, NGOs and international associations are the second most important actor (e.g., 33% of the total area treated is due to their efforts to mobilize funds).

Once mobilized, these funds dedicated to DLRA are put into action by six categories of operators (Table S3). Their contribution varies (Figure 9). Again, in this role, the Government of Niger is the main actor. It has implemented actions covering 63% of the total area treated. The remaining third was made possible by the action of the UN agencies (18%) and the NGOs/international associations (19%). The action of other operators (e.g., national NGOs/associations, sub-regional bodies, and municipalities) constituted less than 4% of the LADA. Development programmes and projects have been the most effective vehicles to put DLRAs into action (Table S4 and Figure 9). The largest areas of DLRA, amounting to 149,137 ha (53%), were implemented under national development projects (e.g., the Climate-Sensitive Agriculture Support Project, the Resilience Building Project to Combat Food Insecurity in Niger, and the SLM Project). International development programmes and projects (e.g., World Food Programme and Action Against Hunger) contributed 19% and 11% of the DLRAs, respectively. In our database, national development programmes and sub-regional projects have very little involvement in SLM actions.

At the very end of the chain of actors (Figure 9), national development projects (e.g., the Community Action Project for Climate Resilience and the Emergency Project in Support of Food Security and Rural Development) were the main contributors to the implementation of the DLRAs (31%) on the ground (Table S6). In contrast, sub-regional development projects, international development cooperation agencies, and local authorities contributed very little to field interventions (merely 1%). The other modalities carried out between 18% and 19% of the total area of DLRAs.

Figure 9 points out the key role of the State of Niger as a fund mobilizer and an operator, supporting more than 60% of the effort in the fight against land degradation in Niger.

## 4. Discussion

### 4.1. Traceability and Sharing of DLRA Data in Niger

An effective SLM policy requires a comprehensive and spatially based assessment of the current situation [26]. Our study revealed the existence of a few non-operational sectoral DLRA databases [20,27,28]. Most of the available data on DLRA suffer from a lack of accessibility (most often not organized in a table or database), reliability, and geolocation. However, the vast majority of data are used by some actors to construct indicator tables, graphs, or maps [28,29]. However, this use is limited to each of the actors, without sharing or archiving according to common criteria [20,28,30,31]. Although useful for each of the actors, this fragmentation of knowledge is not up to the challenges, on a national scale, of monitoring and evaluating actions and their impacts, avoiding the repetition of what has already been done, and the necessary coordination of the various actors in their respective roles. This situation is not an asset for policy making [32,33] or practices to ensure the proper monitoring and assessment of land degradation in Niger [20,34].

The effective implementation of a geolocalized data table that allows the traceability of DLRAs is a guarantee of better visibility of actions in the field [35,36] and a better

evaluation of the use of SLM practices and the sustainability of project effects [37]. Its value lies in the fact that it can prove valuable in improving knowledge [38], especially if it is used and interpreted within a scientifically rigorous methodological framework [39]. Such a data table would facilitate communication between partners and stakeholders in the development of SLM strategies, programmes and projects [40]. These interactions are also a guarantee of good cooperation between all actors. Their respective actions could be based on the same body of data [25].

In this perspective, the need to access spatially referenced databases with a common architecture has been demonstrated [25] according to a harmonized nomenclature. The choice to build a single data table in this work is based on the absence of such a database. The collection of available data, which are scattered, of heterogeneous quality, and of very different natures and formats [41,42], and then formatting and harmonizing them, was therefore a prerequisite for their processing [43,44]. Based on criteria useful for their traceability, it was possible to set up a national data table integrating both information on geolocated DLRAs and information on the network of actors involved. The national data table, which is probably incomplete, nevertheless has the potential to be a useful source of information for the formulation of recommendations to decision makers in the implementation of strategies for capitalizing on and scaling up DLRA.

The method of construction as well as the final structure of this Niger data table could constitute a replicable model in other countries of the GGW zone. In the future, it can help the various data providers to adapt their own data tables (structure and nomenclature) to feed (automatically or not) the national data table more easily. However, ways must be found to maintain and feed such a spatially referenced data table at the national level in the long term for the spatiotemporal monitoring and assessment [25,28,29,45] of DLRA. One way to do this would be to strengthen the skills of the structures that produce these data so that they can maintain such a database, and then build a central national web portal (or even a regional one in the context of initiatives such as the GGW). Such a portal would make it possible to query the data tables distributed in the various structures in order to generate the national table automatically (in the sense of using the most up-to-date data). This web portal could also make it possible to visualize the data on maps according to a standardized graphic semiology, and to navigate through the data according to a geographical query.

These perspectives raise a number of questions: What is data governance? Which structures should produce the data along the chain of actors, from donors to the entity that executes DLRA on the ground? Where to host such a centralized service? What steps should be taken to ensure that the distributed data tables are compatible with the query on the central service? How can the membership of data-producing structures be increased? etc. If each GGW country were to organize such spatially referenced DLRA data systems, this could help, for example, to feed the regional SIOBAP system (System of Information, Observation, Early Warning and Response) that the Pan-African GGW Agency (PAGGW) is setting up.

### 4.2. DLRA Adapted to the Needs of the Population and Administrative Regions of Niger

The analysis of the results of the geolocated data table revealed a wide diversity of DLRAs conducted. These results are in line with those of many works carried out in Sahelian environments [6,46–51]. Indeed, each of these techniques is adapted to a specific socio-ecological situation. Among this diversity of DLRAs, our results showed that dug-out structures are the most frequent. These include soil and water conservation and soil defence and restoration work (SWC/SDR). They are adapted to Sahelian and Sudano-Sahelian zones [52]. Several studies have shown the effectiveness of these techniques in improving the productivity of degraded land [53–56]. They are an effective way to better manage water and reduce land degradation [46], and protect vegetation and biodiversity [49], by increasing and stabilizing agricultural, forestry, and forage yields [6]. Other work has also confirmed that these types of excavated structures increase soil moisture and nutrient

availability [48] and promote crop growth [50,57–59]. These arguments are consistent with the results of Ado et al. [50], who showed that zaï structures, half-moons, which are widespread in Niger where they were first used in the Tahoua region, allow for the growth and development of sorghum crops on land that is initially ridged and uncultivated. These structures make it possible to increase agricultural production [60] and improve the food security of the population [61–63].

Our results showed that geolocated DLRAs spatially occupy more than the southern half of Niger, corresponding to the agrosylvopastoral production zones [64]. They confirm the widely supported conclusions that the areas of SLM intervention in Niger are located on the part of the continent formed by ancient, strongly granitized, and metamorphosed terrains [65].

### 4.3. Diversity of the Network of Actors and Importance of Official Development Assistance in the Implementation of DLRAs in Niger

Funding sources are most often mobilized through multilateral and bilateral mechanisms involving technical and financial partners (TFPs) and NGOs [66]. Moreover, it is not uncommon for technical and financial partners to implement their projects and programmes through their own structures and using their own expertise [67].

Our analysis in Niger sheds particular light on the diversity of actors with specific roles. Some of them, such as the State of Niger, play several roles: as a funder, fundraiser, implementing agency, and executor through its own programmes. The State of Niger complements its reduced capacity to finance from its own funds by a very strong mobilization of funds from donors [27,31,67].

Our results also showed that local governments play an important role in SLM in Niger. This is due to the fact that they mobilize a lot of additional funding through twinning with other communes in developed countries or through decentralized cooperation [68–71]. These additional resources for SLM are implemented through medium- and long-term projects. They also make co-financing contributions, often quite substantial, for SLM actions carried out by development projects or by NGOs. Local authorities, in particular communes, provide additional funding by including in their communal development plans and annual investment plans the financing of actions to combat land degradation [67].

Given certain administrative difficulties, many donors have chosen to work directly and exclusively with NGOs and associations. The size of the envelope devoted to SLM work, coupled with this change in policy, has encouraged the emergence of NGOs involved in SLM. Indeed, most of them, especially national ones, mobilize and execute contracts and agreements with national projects and programmes. They constitute a substantial funding force for SLM, as they are able to mobilize and implement funds that other actors do not or no longer have access to. Within the framework of the "do-it-yourself" approach, several funding options exist and are within the reach of local NGOs. In many cases, they lobby and advocate to mobilize funds other than those of national projects and programmes and intervene in the form of real implementing agencies [67,72]. They themselves call on internal expertise and implement many externally funded projects that are not included in the government's investment budget [66]. Additionally, several NGOs and associations are involved in promoting proven SLM practices. On the ground, NGOs are present, especially in food-insecure areas, as they benefit from significant resource allocations [25].

### 5. Conclusions

Reducing and slowing down land degradation, and rehabilitating or restoring degraded land are key levers in achieving sustainable development for the benefit of populations—particularly those whose livelihood relies on ecosystem services. Combatting land desertification has been declared a national priority in the Niger, echoing target 15.3 of the SDGs. Based on a collection of public data held by different entities, we have built a unique database consisting only of georeferenced data, containing all the information not only on DLRA, but also on the actors and their roles as donors, fund mobilizers, operators,

and implementers. This is a clear choice based on the objective of our work. We are aware that not all field actions are itemized. However, our work demonstrates the added value of creating such a georeferenced database management system in order to (i) deploy targeted sustainable land management initiatives that complement both past and ongoing actions (thus avoiding multiple actions in the same place) and (ii) synergize all the stakeholders. Nevertheless, the task of identifying, harmonizing/standardizing, and geolocalizing DL-RAs on a country-wide scale is considerable. The work in the Niger is unfinished but offers a roadmap towards consolidating these achievements and ensuring their transferability to other countries. The creation and continuous feeding of a database management system must also be undertaken. To this end, raising awareness and mobilizing all stakeholders to contribute collaboratively to such a common dataset represents a major challenge.

However, the efforts involved in such an undertaking are small compared to the benefits of acquiring such a database. It will help in assessing and monitoring the contribution of DLRAs to the SDGs, and thus give value and visibility to the role of each stakeholder. The establishment of scientific observatories anchored in the territories can meet this ambition.

**Supplementary Materials:** The following supporting information can be downloaded at: https://www.mdpi.com/article/10.3390/land12051064/s1.

**Author Contributions:** Conceptualization, J.-L.C., M.L., N.S.J., A.-S.S. and H.B.-A.I.; methodology, A.-S.S., J.-L.C., M.L., N.S.J. and H.B.-A.I.; software, A.-S.S., J.-L.C. and M.L.; validation, J.-L.C., M.L., N.S.J., A.-S.S. and H.B.-A.I.; formal analysis, J.-L.C. and A.-S.S.; investigation, J.-L.C., M.L., N.S.J., A.-S.S. and H.B.-A.I.; data curation, A.-S.S. and J.-L.C., writing—original draft preparation, A.-S.S.; writing—review and editing, J.-L.C., M.L., N.S.J.; and H.B.-A.I.; visualization, J.-L.C., A.-S.S. and M.L.; supervision, J.-L.C.; project administration, J.-L.C., funding acquisition for IRD, J.-L.C. All authors have read and agreed to the published version of the manuscript.

**Funding:** This research was mainly funded by GEF of funder (Grant number 9825). Additional funding was provided by the IRD under seed fund for Great Green Wall studies.

**Data Availability Statement:** The data presented in this study are public and available on request from the corresponding author. The data are not publicly available due to the fact that all have not been published.

**Conflicts of Interest:** The authors declare no conflict of interest.

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
