# Peer review of "Standardized Description of Degraded Land Reclamation Actions and Mapping of Actors’ Roles: A Key Step for Action in Combatting Desertification (Niger)"

_land, doi:10.3390/land12051064_

Round 1

Reviewer 1 Report

This paper introduces the practical experience of managing land degradation in Niger, which is a meaningful choice of topic. However, there are some deficiencies in the structure and content of this paper. I would suggest the following:

1. The conclusion section is missing from this paper.

2. The title includes "the complexity of the network of actors", but in the result section, I did not find any information about complexity or complex network.

3. This paper only lists the spatial layout and input of these projects, which is not enough to support the title of this paper. What impact have these projects had on the local ecosystem?

4. Figure 9 is difficult to understand and needs additional explanation.

5. Some papers can provide reference for revising the paper: https://doi.org/10.1016/j.agee.2021.107757;https://doi.org/10.1111/rec.13317.

Author Response

We thank the reviewers for the work done on the article entitled “The complexity of the network of actors and diversity of sustainable land management actions: the case of Niger”.

We have taken into account all the recommendations made to us to improve this publication which has been revised extensively

Reviewer 2 Report

The manuscript could benefit from language editing.

Round 2

Reviewer 1 Report

The authors have revised the paper, can be accepted